# Self-Supervised Encoders Are Better Transfer Learners in Remote Sensing Applications

Zachary D. Calhoun [1,*], Saad Lahrichi [2], Simiao Ren [3], Jordan M. Malof [4] and Kyle Bradbury [3,5]

1 Department of Civil and Environmental Engineering, Duke University, 121 Hudson Hall, Science Dr, Durham, NC 27708, USA

2 Division of Natural and Applied Sciences, Duke Kunshan University, No. 8 Duke Ave., Kunshan 215316, China

3 Department of Electrical and Computer Engineering, Duke University, Durham, NC 27708, USA

4 Department of Computer Science, University of Montana, Missoula, MT 59812, USA

5 Nicholas Institute for Energy, Environment & Sustainability, Duke University, Durham, NC 27708, USA

* Correspondence: zachary.calhoun@duke.edu

**Abstract:** Transfer learning has been shown to be an effective method for achieving high-performance models when applying deep learning to remote sensing data. Recent research has demonstrated that representations learned through self-supervision transfer better than representations learned on supervised classification tasks. However, little research has focused explicitly on applying self-supervised encoders to remote sensing tasks. Using three diverse remote sensing datasets, we compared the performance of encoders pre-trained through both supervision and self-supervision on ImageNet, then fine-tuned on a final remote sensing task. Furthermore, we explored whether performance benefited from further pre-training on remote sensing data. Our experiments used SwAV due to its comparably lower computational requirements, as this method would prove most easily replicable by practitioners. We show that an encoder pre-trained on ImageNet using self-supervision transfers better than one pre-trained using supervision on three diverse remote sensing applications. Moreover, self-supervision on the target data alone as a pre-training step seldom boosts performance beyond this transferred encoder. We attribute this inefficacy to the lower diversity and size of remote sensing datasets, compared to ImageNet. In conclusion, we recommend that researchers use self-supervised representations for transfer learning on remote sensing data and that future research should focus on ways to increase performance further using self-supervision.

**Keywords:** machine learning; self-supervision; computer vision; transfer learning; SwAV; semantic segmentation; satellite imagery; domain adaptation

## 1. Introduction

Deep Neural Networks (DNNs) typically require large labeled training datasets to achieve satisfactory recognition accuracy. This is a major limitation of DNNs in remote sensing applications, where the data are often limited in availability, expensive to purchase, and time-consuming to label. In object recognition applications, for example (e.g., detection of aircraft, boats, oil rigs), large quantities of overhead imagery must be manually inspected to find instances of the target objects, and there is no guarantee that a sufficiently large number of target instances will be found in any particular collection of overhead imagery. Due to these challenges, training data are often scarce in remote sensing applications, which limits the recognition accuracy of DNNs. This problem is often referred to as a few-shot or low-shot learning problem, and it is a well-studied problem in machine learning [1–4].

Several approaches have been proposed to address this problem. One strategy, called data augmentation, increases the dataset size by applying transformations to the original dataset that allow each image to be used multiple times in the model (e.g., by rotating an image 90%, each image has four distinct views possible) [5]. Another strategy, synthetic

imagery, consists of generating data using simulated overhead imagery [6]. Both data augmentation and synthetic imagery are methods to increase the amount of training data artificially. A complementary approach is to pre-train a model so that it does not learn from scratch but rather learns low-level features like edges and shapes that would be shared between datasets. This strategy is called transfer learning, and it is a standard method for model initialization as it can improve accuracy and decrease training time [7]. An example transfer learning pipeline can be visualized in Figure 1.

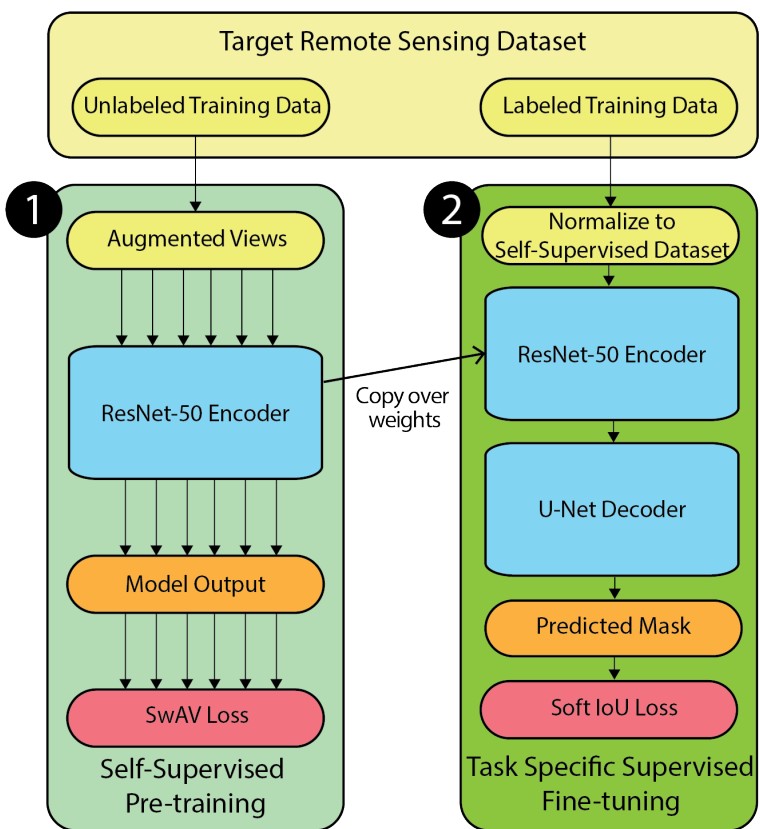

**Figure 1.** The model architectures used in our experiments. In the self-supervised pre-training step (1), the training data were fed through a ResNet-50 architecture, and the SwAV loss from the fully connected layer was used to train the model. Once the ResNet-50 encoder was trained, the fully connected layer was detached, and the model weights were saved. (2) To fine-tune, the ResNet-50 model was attached to a U-Net decoder, with the ResNet-50 weights coming from the previous step and the remaining weights randomly initialized. During training, both the U-Net and ResNet-50 weights were updated using the soft-IoU loss between the actual and the predicted masks.

The efficacy of transfer learning depends on the pre-training dataset and the pre-training task. In theory, the pre-training dataset should be similar to the target dataset. For example, if one wanted to train a model to detect airplanes in satellite images, it would make intuitive sense that a model pre-trained to detect cars in satellite images would outperform a model pre-trained to detect dogs in ground-level captured natural imagery, as the large satellite imagery dataset would help the model learn features that are shared with the target dataset. However, this is not necessarily the case. Pires de Lima et al. [8] showed that ImageNet, a large natural imagery dataset, frequently outperforms remote sensing datasets when used in a supervised classification pre-training task, even when the target dataset is in the remote sensing domain. This is because ImageNet contains 1000 classes and 1.4 million images. The dataset size and class diversity allow the model to learn a variety of features that are unmatched by smaller remote sensing datasets [9]. Furthermore, the optimal model weights for the supervised ImageNet classification task are

freely available through the most commonly used machine learning frameworks, making this pre-trained model a popular choice among practitioners.

Regarding the impact of pre-training tasks on transfer learning, recent research has shown that self-supervised methods often outperform supervised methods [10]. For example, self-supervision applied to the ImageNet dataset outperforms supervised classification on ImageNet when applied to remote sensing tasks [11]. Self-supervision involves using a pre-training task that does not require explicitly labeled data to uncover features that might benefit the model on target tasks. Instead, implicit labels are derived from the unlabeled data themselves. For example, an image might be randomly rotated, and the self-supervised task would be to predict the amount by which the image has been rotated (i.e., the classes would be 0, 90, 180, and 270 degrees). Through this task, the model may learn which features determine the intrinsic orientation of the images. Other examples of such tasks include colorization, prediction of missing pixels, and solving jigsaw puzzles [12–15]. Self-supervised methods enable the model to learn features beyond those learned through supervision, as supervised methods merely teach the model to learn those features that help the model discriminate between the manually labeled classes. In contrast, self-supervised methods teach the model to learn inherent features of the dataset, which, depending on the method, may allow for more expressive features to be learned.

Despite self-supervision's promise, many state-of-the-art self-supervision methods today are often difficult to implement in practice. They require large datasets and substantial computational resources to be effectively applied, which precludes their widespread use among practitioners. For example, SimCLR and SimCLR v2 [16,17] achieved state-of-the-art performance on the ImageNet classification task using batch sizes of 4096, while another method, BYOL [18], achieved state-of-the-art performance, but required 64 graphics processing units (GPUs). Access to this number of GPUs may limit users of this method to those with access to high-performance computing environments or to those willing to pay for additional resources. Because of this additional hurdle, the efficacy of self-supervised methods is dependent upon their ability to pre-train models that transfer well to multiple domains. Essentially, we want to apply self-supervised pre-training once to a large dataset and reuse the learned weights in various of circumstances. To make such a pre-trained model optimized for the domain of remote sensing, a variety of self-supervision techniques have emerged [19–21]. Recent research has suggested that, while these methods result in pre-trained models that transfer better to remote sensing applications than the supervised ImageNet encoder, the self-supervised ImageNet encoder still frequently wins [10,11,22]. Again, this is likely due to ImageNet's size and diversity.

Because the model pre-trained on ImageNet still prevails, Risojevic et al. suggest a paradigm called domain-adaptive pre-training, in which a model is pre-trained on ImageNet, then further pre-trained on a remote sensing dataset so that it can adapt to the remote sensing domain [11]. A similar strategy referred to as hierarchical pre-training by Reed et al. supports this paradigm by showing that self-supervised pre-training is improved when it occurs sequentially over multiple datasets. In our work, we seek to explore this paradigm further using three distinct remote sensing datasets. Previous work on remote sensing datasets was limited to classification tasks, so we opted instead for semantic segmentation tasks, as these tasks are generally more difficult and are relevant to a broader range of remote sensing tasks. Furthermore, we seek to test this paradigm using a less computationally expensive self-supervision method, SwAV [23]. This self-supervision method has demonstrated state-of-the-art performance on the ImageNet classification task and is easily implemented using smaller batch sizes and fewer GPUs, which makes applying SwAV to new datasets cheap and accessible for increased reproducibility. The code for this method is open-source, and pre-trained models are available through the Torch model Hub. A more detailed description of how SwAV works can be found in Appendix A.

By testing the ability of SwAV pre-trained encoders to transfer to remote sensing datasets, we hope to inform practitioners in the field of a more effective and computationally

efficient method for boosting performance on target tasks. Thus, through our experiments, we make the following contributions:

- We further demonstrate that the model pre-trained through self-supervision on ImageNet outperforms the model pre-trained using the supervised ImageNet classification task on three distinct remote sensing tasks, suggesting that practitioners should apply transfer learning using self-supervised models as a standard practice.
- We evaluate the performance improvement of a model pre-trained using SwAV on ImageNet, along with the extra improvement possible through further pre-training, with different amounts of labeled training data available for fine-tuning. We found that using of the self-supervised encoder is most beneficial in data-limited scenarios, and as more labeled data are available for the target task, the performance gap between encoders decreases.
- We show that, when using the SwAV model with pre-trained weights from ImageNet, further pre-training of SwAV on the target dataset does provide additional accuracy improvements, although often these improvements are minor. Our experiments suggest that a larger unlabeled target dataset yields larger improvements during this additional pre-training step.

The code used for our experiments is publicly available on GitHub, linked in the Data Availability Statement.

## 2. Materials and Methods

### 2.1. Experimental Design

Each of our experiments consisted of two steps: (1) the pre-training of an encoder on a specified dataset and (2) the fine-tuning of that encoder on a target task. Since our experiments aimed to evaluate the impact of the pre-training pipeline on performance, we developed five different pre-training pipelines, as outlined in Table 1. We then applied these five pre-trained encoders to three target datasets so that we could compare relative performance in each of the datasets.

**Table 1.** The first experiment used an encoder pre-trained using supervision on the ImageNet classification challenge. The second encoder was pre-trained using SwAV on ImageNet, and the third was pre-trained using SwAV on just the target data. For experiment four, we applied SwAV to the target data using the weights initialized from being pre-trained on ImageNet. Lastly, for experiment five, we took the initialized encoder using SwAV on ImageNet and then further applied SwAV using all three datasets to determine whether the model benefited from seeing a larger quantity of remote sensing data.

| Experiment # | Pre-Training Method | Stage 1 Pre-Training Dataset (via Weight Initialization) | Stage 2 Pre-Training Dataset |
|---|---|---|---|
| 1 | Supervised | ImageNet | None |
| 2 | Self-supervised (SwAV) | ImageNet | None |
| 3 | Self-supervised (SwAV) | None | Target |
| 4 | Self-supervised (SwAV) | ImageNet | Target |
| 5 | Self-supervised (SwAV) | ImageNet | Building + Field Delineation + Solar |

For the first step of the experiment, pre-training consisted of either downloading the weights of a model using Torch's model hub or applying SwAV to a dataset or combination of datasets. The supervised ImageNet and SwAV ImageNet encoders were available on Torch's model hub, making them easy to use in practice. We explain how to download these model weights in Appendix B. When applying SwAV, we used the code written by the SwAV authors and applied it to the target datasets, with one small change: we implemented a flag to indicate whether or not to initialize the model using the SwAV ImageNet weights

before applying it to the specified dataset. Further details on applying SwAV are provided in Appendix C, with the scripts used to create each model available in this project's GitHub. Since both model weights were available for the ResNet-50 architecture, we used this architecture as the encoder for all our experiments.

For the second step of our experiments, we attached the pre-trained ResNet-50 encoder to a U-Net decoder, as visualized in Figure 1 [24,25]. This architecture was chosen because all of the downstream tasks selected were semantic segmentation tasks and because the ResNet encoder to U-Net decoder model has been shown to be remarkably successful within the field of semantic segmentation in remote sensing [26].

Task fine-tuning was performed on an NVIDIA GeForce RTX 3080. A learning rate of 1e-3 with a batch size of 16 was used for all experiments using the Adam optimizer. The soft IoU loss, based on the loss function used in [27], was used for all tasks. For each experiment, we trained the model for 100 epochs. At each epoch, validation set performance was calculated, and final model parameters were selected based on this validation performance. For each task, the training dataset was restricted to 64, 128, 256, 512, and 1024 images. This was so that we could evaluate the impact of pre-training under real-world conditions where practitioners have a varying amount of target labels. Lastly, we performed the fine-tuning step three times for each encoder, task, and dataset size combination so that we could capture a mean and variance for each model.

### 2.2. Datasets

Three semantic segmentation tasks were selected in the domain of remote sensing. These tasks consisted of a building segmentation task, a field delineation task, and a solar photovoltaic array segmentation task. These tasks were chosen because the datasets were sufficiently distinct from one another, which would allow any conclusions to generalize beyond looking at any one task alone. Additionally, semantic segmentation is more challenging than scene classification, meaning that our results might generalize better to other tasks. Table 2 details the datasets, the training and test sizes, the image sizes, and the task type.

The training and test datasets were mutually exclusive for each of the datasets. The training dataset was divided into subsets of 64, 128, 256, 512, and 1024 images so that target task performance could be evaluated using a varying labeled dataset size. The same test set was used for each of the models learned so that performance could be comparatively evaluated. Apart from the smaller training datasets used for fine-tuning, the number of images used for self-supervised pre-training was unrestricted so that all unlabeled data could be used. This extra unlabeled data reflect conditions likely to be found by practitioners, as unlabeled data might be abundant and labeled data scarce.

**Table 2.** The three semantic segmentation datasets used for our experiments. The training size varies from 1572 for the farm parcel delineation dataset to 75,020 for the Inria aerial image labeling dataset, allowing us to study the effect of training dataset size. The images either come in size 224 × 224 or as 5000 × 5000 Sentinel-2 tiles that we retile to get 224 × 224 tiles.

| Dataset | Train Set Size | Validation Set Size | Image Size | Task Type |
|---|---|---|---|---|
| Inria Aerial Image Labeling | 75,020 | 1024 | 5000 × 5000 (retiled to 224 × 224) | Semantic segmentation (pixelwise labeling) |
| Solar Photovoltaic Array Detection | 28,245 | 1024 | 5000 × 5000 (retiled to 224 × 224) | Semantic segmentation (scarce object detection) |
| Farm Parcel Delineation | 1572 | 198 | 224 × 224 | Semantic segmentation (boundaries delineation) |

### 2.2.1. Building Segmentation

The INRIA building segmentation challenge provides a dataset with several sets of images, where each set corresponds to a specific city. For each city, masks are provided that outline buildings in the images [28]. For our experiments, we only used the dataset for Austin, Texas, with the training/validation split done according to the recommendations from the referenced paper. Again, the data were re-tiled to 224 by 224 pixels for use in our experiments. The whole training data were used for the self-supervision step, consisting of 75,020 images.

### 2.2.2. Solar Photovoltaic Array Detection

The solar photovoltaic dataset, further described in [29], provides high resolution (0.3 m/px) satellite imagery of several cities in California with labeled solar arrays. This task serves as an example of scarce object detection within semantic segmentation, in which the object is not seen in every image. However, the object itself is relatively trivial to select when it does appear in the image.

For our experiments, the images were downloaded and split into training and test sets, using an 80/20 split. Since the images were 5000 by 5000 pixels, the images in each set were re-tiled to be 224 by 224 pixels for the ResNet-50 pipeline. That is, we took the original images and divided them into chunks of 224 by 224 images for simpler input into the ResNet-50, which takes this size by default. For self-supervision, we used the entire training dataset (without labels) resulting from the re-tiling, which consisted of 28,245 images. To get the smaller training sets for the fine-tuning step, we randomly sampled 64, 128, 256, 512, and 1024 images from the training dataset.

### 2.2.3. Field Delineation

The field delineation dataset, introduced in [30], uses Sentinel-2 satellite imagery of size $224 \times 224$ pixels at a 10m resolution. This dataset consists of boundaries of farm parcels in France, with the option to configure the mask as the boundary or as the parcels themselves. For our experiments, we used the more challenging task of predicting the boundaries, and we randomly selected 64, 128, 256, 512, and 1024 images from the training set for model training. We used the validation dataset provided by the dataset authors to evaluate performance.

The total size of the training set is 1572 images, all of which were used for self-supervision. This smaller dataset serves as a useful example for illustrating the lower performance of self-supervision when trained using a smaller dataset.

## 3. Results

The results of our experiments are summarized in Figure 2, in which the pixel IoU for each target task is plotted as a function of the fine-tuning dataset size, with each of the pre-training paradigms shown as a separate line. The general ranking, from worst performing to best performing, tends to be SwAV trained on the target data, then the supervised encoder, SwAV trained on ImageNet, then SwAV initialized on ImageNet with in-domain data.

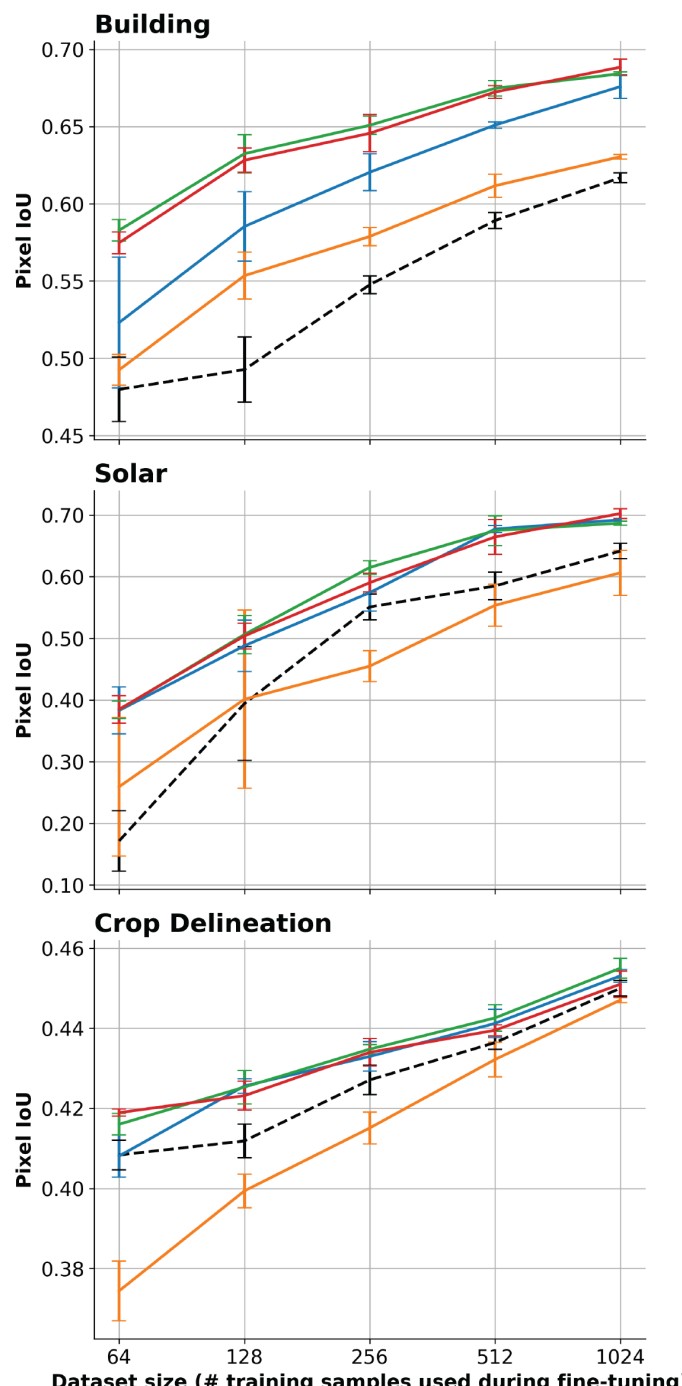

**Figure 2.** Pixel IoU plotted as a function of the number of labeled training samples used in fine-tuning, with results separated out by task.

### 3.1. Representations from Self-Supervision on Imagenet Transfer Better to New Tasks Than Supervised Representations

We demonstrate that empirically, a pre-training strategy that uses SwAV on the ImageNet dataset results in an encoder that transfers better to remote sensing tasks than supervised training on labeled data. The average performance boost resulting from using self-supervision is visualized in Figure 3. The difference is always positive, demonstrating the gains from self-supervision. The difference in performance appears to diminish with a larger set of labeled data, suggesting that the self-supervised encoders may have similar performance to the performance of supervised encoders with larger sets of labeled data. However, we may consider the performance of the self-supervised encoder as an upper bound on the performance of the supervised encoder, suggesting that the self-supervised encoder should be preferred.

We also note that the magnitude of differences is far more significant in the building and solar datasets than the difference found in the crop delineation dataset. Qualitatively, the crop delineation task is a harder segmentation task. The satellite imagery is of much lower resolution, and the borders of fields represent a visual feature that is much more difficult for humans to predict than the border of a building or a solar panel on a roof. Quantitatively, the magnitude of improvement from increasing the dataset sizes supports this claim. The IoU difference between the worst and best-performing model is 0.07 in the crop delineation dataset. In the solar dataset, the difference is 0.6, and in the building dataset, the difference is 0.4. This indicates more room for improvement with the building and solar datasets.

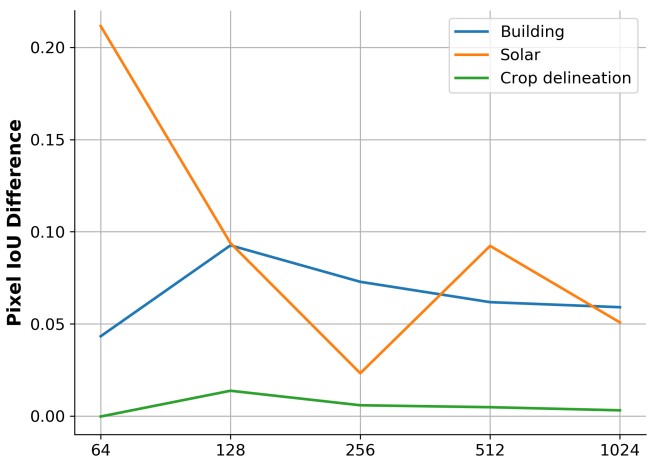

**Figure 3.** Pixel IoU increase resulting from using a self-supervised encoder on ImageNet over the supervised ImageNet encoder.

### 3.2. Further Self-Supervised Pre-Training Is of Limited Benefit

While further pre-training beyond the initial pre-training on ImageNet does appear to boost the performance of the model slightly, it is not clear under which conditions this performance boost might be expected. Again, we summarize the performance boost using the difference in pixel IoU in Figure 4.

Notably, the building dataset benefits the most from further pre-training, which makes sense given the comparatively larger dataset size (the building dataset used for pre-training has three times more data than the solar dataset and seventy times more data than the crop delineation dataset). This benefit disappears when there is sufficient labeled data, suggesting that having more labeled data is still more important than having more unlabeled data.

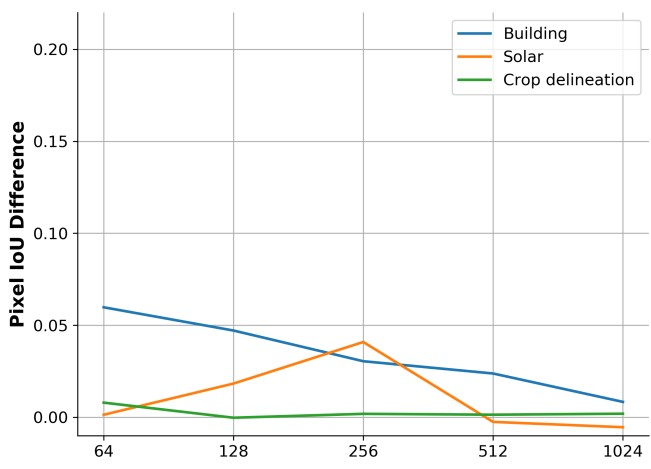

**Figure 4.** Pixel IoU increase resulting from further pre-training on the target dataset over the supervised ImageNet encoder.

### 3.3. The Number of Prototypes Used in Additional Pre-Training Should Match the Original Model'S

In the original SwAV paper [23], the authors trained SwAV using 3000 prototypes and found that increasing that number leads to negligible performance gain, given the computational cost. To analyze the impact of the number of clusters in further pre-training, we used the original pre-trained SwAV model and further pre-trained it using 100, 1000, 3000, and 5000 prototypes on the INRIA building segmentation dataset. Our results in Figure 5 show that the best-performing model is the one matching the number of prototypes used during initial pre-training. The performance gap was most noticeable when using a small data size. When using 1024 images during fine-tuning, the performance gap is negligible. This suggests that better results are achieved when the number of prototypes used in further pre-training matches the number of prototypes in the original model (in this case, 3000 were used for ImageNet pre-training). Further research may explore the impact of this relationship number by also varying the number of prototypes in the original model and re-testing this hypothesis. Given the results, our recommendation for practitioners under similar situations is to keep the same number of prototypes during target-side pre-training.

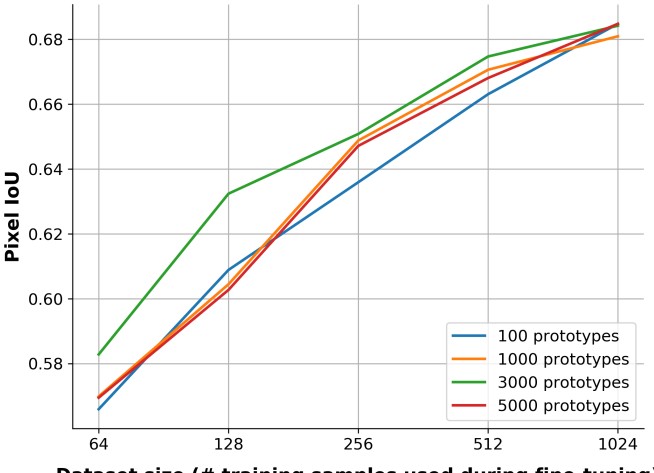

**Figure 5.** Pixel IoU plotted against dataset size for the INRIA building segmentation task. Additional pre-training using 3000 prototypes, matching the original model, outperforms all the other models (green line), although the gap is hardly noticeable as the data size increases.

## 4. Discussion

Our experiments demonstrate that the representation learned through self-supervision on ImageNet transfers better to remote sensing applications than the representation learned through the supervised ImageNet classification task. This conclusion has been suggested in recent work [10,11,31]. However, to our knowledge, this work is the first to focus explicitly on applying a hierarchical self-supervised pre-training paradigm to remote sensing. Furthermore, we strengthen the conclusions of previous work by conducting our experiments on semantic segmentation tasks. Our work suggests that the previous standard of initializing a model using the supervised ImageNet encoder could be replaced with the self-supervised encoder for better performance at no extra computational cost, as these self-supervised encoders are freely available and easily accessible.

We additionally confirm the finding that self-supervised encoders pre-trained on ImageNet tend to transfer particularly well, as discussed in [8]. This finding is counter-intuitive, as one might expect that a pre-training strategy that uses remote sensing data would result in better performance on remote sensing tasks. Previous work has explored the underlying reasons why ImageNet tends to transfer so well, with one main finding being that it has many classes and that each class is well represented with lots of data [32]. ImageNet has over 1 million images spread over 1000 classes, whereas our largest dataset only has about 75,000 images and no explicit classes. Indeed, we suspect that the better performance of SwAV on the building task can at least be partially attributed to its larger dataset size. We suggest that future work should continue to compare the performance of the self-supervised ImageNet encoder to self-supervised encoders trained on remote sensing datasets to continue the search for a domain-specific general encoder.

Furthermore, our work confirms that additional pre-training on the target data, initialized using SwAV pre-training on ImageNet, improves performance, as suggested in [22]. We emphasize that our proposed paradigm is novel from these past works in that we suggest initialization using SwAV on ImageNet (which takes no extra computational effort than initialization using supervised ImageNet weights) and optional further pre-training using SwAV on the unlabeled target data (which avoids additional data collection). Furthermore, our work focuses on performance within the domain of remote sensing.

We additionally find that, in the case of SwAV, the initialization with ImageNet self-supervised weights prevented model collapse when applying SwAV to the target data. As the original SwAV paper discussed, mode collapse may sometimes occur during training, in which all images are assigned to the same cluster (or prototype). This occurred on the crop delineation dataset when not initialized using ImageNet weights but did not occur otherwise. To fix this problem in our experiments, we adjusted the hyperparameters following the suggestions made by Caron et al. in the code repository on GitHub [23].

One limitation of our experiment was the fine-tuning of SwAV hyperparameters to work better with remote sensing data. Specifically, we suspect that the multi-crop augmentation strategy might not be optimal for remote sensing imagery. This augmentation strategy applies two rounds of crops to the 224 by 224 pixels images, then re-scales the augmented images to be 224 by 224 pixels. Because remote sensing images often comprise multiple objects of interest, as opposed to natural imagery, which usually contains one subject, we suspect that the multi-crop strategy may not always be cropped in such a way to capture related objects. As such, the resulting cluster assignments would be expected to be far away. If the target imagery is high resolution and the dataset is full of spatially close objects, then this multi-crop strategy would make sense, as is the case with the building dataset. Because a building is likely to be a large structure captured in multiple crops, the cluster assignments would capture similar features of that building. This would not be the case with the solar dataset, in which solar arrays are sparsely populated throughout the dataset, and the images contain a mix of suburban and rural areas.

Future work may address these limitations and improve overall performance. For example, continued optimization of the SwAV hyperparameters and network configuration optimal for transfer learning to remote sensing tasks may demonstrate further performance

gains. While we used a convolutional neural network backbone, an alternative, the vision transformer (ViT) backbone, has been found to explicitly encode spatial information when trained using self-supervision, which may result in an improved ability to perform transfer learning [33].

Other training datasets may also yield performance improvements including pre-training based on the more diverse ImageNet-21K dataset [34]. Alternatively, since our results demonstrate that in some cases, further pre-training on remote sensing data does result in boosted performance, this suggests that a general remote sensing encoder might exist. We believe that the creation of a large, curated remote sensing dataset for training such an encoder, may enable the more rapid development of high-performance remote sensing models.

## 5. Conclusions

This paper presents a new pre-training paradigm that combines the benefits of domain adaptive pre-training and hierarchical self-supervised pre-training [11,22]. In this new paradigm, the encoder is always initialized to the self-supervised SwAV-ImageNet weights. SwAV is then applied to the target dataset to further pre-train using self-supervised learning, which requires no labeled data. Once the pre-training steps are complete, we fine-tune the model using a small number of labeled examples. Across all of our experiments, we show that the self-supervised ImageNet weights consistently transfer better (i.e., yield improved performance) than a supervised ImageNet model for remote sensing tasks. Our proposed transfer learning strategy should provide practitioners with an easy method for achieving better model performance using transfer learning: initialize models with self-supervised weights. Lastly, we show that this benefit mainly depends on labeled and unlabeled dataset sizes. If unlabeled data are abundant, then further performance gains can be made through the additional application of SwAV on the target dataset. As the amount of labeled data grows, however, then the performance gap between paradigms decreases.

In summary, our key recommendation is that computer vision practitioners in remote sensing should initialize deep neural networks through self-supervised pre-training rather than supervised pre-training. This approach can be taken to achieve generally equal or greater performance on the target task without increasing training time, code complexity, or cost.

**Author Contributions:** Conceptualization, J.M.M. and K.B.; methodology, Z.D.C.; software, Z.D.C. and S.L.; validation, J.M.M. and K.B.; formal analysis, Z.D.C.; investigation, S.R. and Z.D.C.; resources, K.B.; data curation, Z.D.C. and S.L.; writing—original draft preparation, Z.D.C.; writing—review and editing, K.B., J.M.M., S.L. and S.R.; visualization, S.L.; supervision, K.B. and J.M.M.; project administration, K.B.; funding acquisition, K.B. All authors have read and agreed to the published version of the manuscript.

**Funding:** This research was funded by the Duke University Bass Connections, Climate+, and Data+ programs.

**Data Availability Statement:** The code for our experiments is split into two repositories. To run SwAV, we forked the original SwAV repository, which can be found here: https://github.com/zcalhoun/swav, (accessed on 30 September 2022). For the experiments applying the encoder for the fine-tuning step, the code is available at https://github.com/zcalhoun/ssrs, (accessed on 30 September 2022).

**Acknowledgments:** We thank the Duke University Climate+ and Bass Connections programs for their financial support and thank Edrian Liao and Rebecca Lan for their collaboration and numerous contributions to this project.

**Conflicts of Interest:** The authors declare no conflict of interest.

## Appendix A. More Details on SwAV

SwAV stands for "Swapping Assignments between multiple Views of the same image", which hints at how the method works [23]. First multiple data augmentations are randomly applied to an input image at training time to produce multiple views of that image. These data augmentations start with a multi-crop strategy, in which the image is cropped to produce two smaller images, then each of those smaller images is further cropped into three images, for a total of 6 views of each image. Each of these multiple views of the input image then has horizontal flipping or Gaussian blur applied to augment the image further.

Following data augmentation, each view (i.e., small image crop) is fed through the ResNet-50 encoder to produce a feature representation of that image. This feature representation is used to define a cluster assignment (also called a prototype in the paper). The distance between cluster assignments between views is then used as the loss function to be minimized. In effect, this loss function encourages the model to assign the same cluster to each view of the same image, which directs the model to learn features that the views have in common during the training process.

The number of clusters is a hyper-parameter defined at model initialization (prior to any pre-training). However, in practice, the number of clusters should be much greater than the number of natural classes in the dataset so that sub-classes within each class may be learned (even if they are not explicit). For example, ImageNet might contain a class "automobile", and we want to ensure that variability within this class is captured, so sub-classes would be sedan, SUV, truck, et cetera. In the original paper, the authors trained their model on ImageNet (which has 1000 classes) using 3000 prototypes.

## Appendix B. Initializing a Model for Transfer Learning

The Torch model Hub (accessible at https://pytorch.org/hub/, accessed on 30 September 2022) is a repository for downloading and publishing deep learning models. Given the popularity of the ResNet-50 architecture, there are plenty of pre-trained model weights to choose from that can be loaded to initialize this architecture. The original authors of the SwAV paper published the model weights for their ResNet-50 model that achieved the best performance with 3000 prototypes, and this is the model we used for our experiments. This model can be loaded through the Torch model hub by loading the ResNet-50 with address `facebookresearch/swav:main`). Before the loaded model weights can be applied to a U-Net architecture, two modifications to the model architecture were needed. First, the model loaded through the Torch Hub comes with a final, fully connected layer. This layer is not used in our architecture, so these weights were discarded. Second, because we used a U-Net decoder, intermediate layer outputs were required from the encoder. We defined a custom version of the ResNet-50 in our GitHub repository that returned these intermediate layers during the forward call. We direct the reader to our GitHub repository to see how these requirements were implemented (https://github.com/zcalhoun/ssrs, accessed on 30 September 2022).

## Appendix C. Further Details on Applying SwAV

For the most part, we copied the SwAV hyperparameters used in the original paper to build the final model. The specific hyperparameters can be seen in the scripts found in our forked version of the SwAV repository (https://github.com/zcalhoun/swav, accessed on 30 September 2022). Of note is that we used 3000 prototypes for each of our experiments for consistency. Additionally, we ran SwAV for 800 epochs for each of the encoders generated in our experiments, following the epochs used in the original paper. We also applied the multi-crop strategy using 160-by-160 pixel crops, then 96 by 96 pixel crops. This was done because our input data were already cropped to 224-by-224 pixels, and thus initial cropping of 224-by-224 would not be effective. Lastly, we ran our experiments using 8 GPUs at a time with batch sizes of 128, to give an effective batch size of 1024.

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
