# Peer review of "Self-Supervised Encoders Are Better Transfer Learners in Remote Sensing Applications"

_remotesensing, doi:10.3390/rs14215500_

Round 1

Reviewer 1 Report

This work demonstrates that the representation learned through self-supervision on ImageNet transfers better to remote sensing applications than the representation learned through the supervised ImageNet classification task. The conclusion is of sufficient interest to the remote sensing community. Before the acceptance, I have one question.

What if we have enough annotation for the supervised remote sensing tasks, does the statement in this work still hold? Have you ever analyzed the number of training samples in a supervised paradigm?

Author Response

We thank the reviewer for their time and feedback.

1.1: What if we have enough annotation for the supervised remote sensing tasks, does the statement in this work still hold? Have you ever analyzed the number of training samples in a supervised paradigm?

RESPONSE: Our experiments suggest that the performance gap decreases as the labeled annotation size increases (as shown in Figures 3-4), and so we should expect that, given sufficient data, the pre-training strategy becomes less critical for performance. However, knowing when there is “sufficient” data is difficult to know a priori in practice. Therefore, we suggest that models be initialized using self-supervision, as it is unlikely to hurt performance, but if the labeled data corpus is sufficiently small, it may improve performance. In the event of sufficient annotated data, the added benefit of using our approach is that models should still converge more quickly as the model will be starting with improved feature representation within the model.

Reviewer 2 Report

This is an excellent manuscript. The proposed method is simple but very powerful. Although hyperparameters such as learning rate, batch size, and optimizer were fixed, the effect might be slight. The results obtained from the experiments can be applied to not only remote sensing but also to a variety of computer vision tasks. If this approach could be applied to ViT backbones, it would further expand the range of applications. Moreover, better results can be obtained if ImageNet-21K is used in the proposed method. This research will open a new door for SSL. Source codes for the proposed methodology are desired to be made available on GitHub or other repositories.

Author Response

2.1 This is an excellent manuscript. The proposed method is simple but very powerful. Although hyperparameters such as learning rate, batch size, and optimizer were fixed, the effect might be slight. The results obtained from the experiments can be applied to not only remote sensing but also to a variety of computer vision tasks. 

RESPONSE: Thank you for your feedback and your time. We agree that further hyperparameter tuning may improve the performance of our models, and we altered our final paragraph in the Discussion section to explicitly mention this as potential future work.

2.2 If this approach could be applied to ViT backbones, it would further expand the range of applications. 

RESPONSE: This is an interesting proposal, and we believe that future work should explore how well our methodology works with non-convolutional backbones. We added a sentence in the last paragraph of our Discussion section to suggest this for future work.

2.3 Moreover, better results can be obtained if ImageNet-21K is used in the proposed method. This research will open a new door for SSL. 

RESPONSE: This is also a good idea. We added a recommendation in the Discussion section to explore how the initial training task impacts transferability.

2.4 Source codes for the proposed methodology are desired to be made available on GitHub or other repositories.

RESPONSE: We link to our GitHub in the Data Availability Statement section at the end of our manuscript, and we further reference this at the end of the Introduction. All the source code needed to reproduce the work are included in that repository.

Reviewer 3 Report

Well written manuscript and concise abstract. Mention past study that follow similar pre training methodology. Line 56 'to' could be removed. Line 163 could b paraphrased to replace the word 'harder task'. What were the criteria for the dataset selection and the specific task selection? 

Author Response

3.1 Well written manuscript and concise abstract. 

RESPONSE: Thank you for your feedback and your time.

3.2 Mention past study that follow similar pre training methodology.

RESPONSE: We reference Pires de Lima et al., Risojevic et al., and Ericsson et al. in the manuscript on lines 41-71, which provides three recent examples of similar methodologies.

3.3 Line 56 'to' could be removed. 

RESPONSE: Thank you for catching this error. We have removed this word.

3.4 Line 163 could b paraphrased to replace the word 'harder task'. 

RESPONSE: Thank you for the recommendation, we have rephrased this statement to be “semantic segmentation is more challenging than scene classification.”

3.5 What were the criteria for the dataset selection and the specific task selection? 

RESPONSE: We selected semantic segmentation because we wanted to focus on particularly challenging tasks that require deep understanding of the content of the image and pixel-wise decisions that often require an understanding of the larger context of the image. Additionally, there were numerous well-studied benchmarks to use for performance evaluation. As for the specific datasets selected, we looked for three distinct segmentation tasks. The building segmentation task is for high resolution (30cm/px) segmentation of a commonly-occurring object. In other words, a majority of the images of populated regions have buildings visible in them. The solar photovoltaic segmentation task is again high resolution satellite imagery, but for a rare object, with solar panels only in 50% of the training images, and generally far less present in the wider population. The final task, crop delineation, uses low resolution imagery (10 m/px), and the task is notably more difficult due to the image resolution. We believe that these distinctions are sufficiently different to be representative of common datasets used in the field and to provide sufficient evidence to support the conclusions of this manuscript.